# The Advancement of Nanomaterials for the Detection of Hepatitis B Virus and Hepatitis C Virus

**DOI:** 10.3390/molecules28207201

**Published:** 2023-10-21

**Authors:** Wanting Shi, Kang Li, Yonghong Zhang

**Affiliations:** 1Interventional Therapy Center of Liver Disease, Beijing You’An Hospital, Capital Medical University, Beijing 100069, China; 17854239722@163.com; 2Biomedical Information Center, Beijing You’An Hospital, Capital Medical University, Beijing 100069, China

**Keywords:** hepatitis B virus (HBV), hepatitis C virus (HCV), detection, nanomaterials

## Abstract

Viral hepatitis is a global health concern mostly caused by hepatitis B virus (HBV) and hepatitis C virus (HCV). The late diagnosis and delayed treatment of HBV and HCV infections can cause irreversible liver damage and the occurrence of cirrhosis and hepatocellular carcinoma. Detecting the presence and activity of HBV and HCV is the cornerstone of the diagnosis and management of related diseases. However, the traditional method shows limitations. The utilization of nanomaterials has been of great significance in the advancement of virus detection technologies due to their unique mechanical, electrical, and optical properties. Here, we categorized and illustrated the novel approaches used for the diagnosis of HBV and HCV.

## 1. Introduction

Viral hepatitis is a global health concern, with 325 million people living with it. It caused 1.34 million deaths in 2015 [1], mostly from hepatitis B virus (HBV) and hepatitis C virus (HCV). HBV and HCV resulted in 1.1 million deaths in 2020 [2]. Globally, 2 billion people are infected with HBV [3] and approximately 58 million people live with a chronic hepatitis C infection [4]. For the 296 million people infected with HBV chronically, only approximately 10% were diagnosed [5]. Besides acute hepatitis and acute liver failure, the major life-threatening event for these patients is the occurrence of liver cirrhosis and oncogenesis [6,7,8]. Early diagnosis and proper management are of great significance.

The diagnosis of HBV and HCV infections primarily relies on serologic testing to detect antibodies, as well as nucleic acid testing (NAT) to detect viral nucleic acid directly. Although enzyme-linked immunosorbent assay (ELISA) and polymerase chain reaction (PCR)-based methods have been commonly utilized to detect HBV, they have limitations such as an insufficient sensitivity for low-level viremia patients, the long time required for the results, and the need for large-scale equipment [9]. Detection methods with a high sensitivity and high specificity for achieving rapid and portable HBV and HCV detection are still in demand.

Advancements in nanotechnologies have facilitated the research of ultra-sensitive and economical biosensors to detect various pathogens, including HBV and HCV. Nanomaterials offer outstanding properties such as a high surface-to-volume ratio, unique electrochemical attributes, and high catalytic activities, which allows for an enhanced sensitivity and selectivity in detecting viruses. Various nanomaterials, such as nanoparticles, nanowires, nanotubes, graphene derivatives, and nanocomposites, have demonstrated their feasibility for virus detection by enabling rapid, highly sensitive, and cost-effective methods [10]. Nanomaterials can be functionalized with specific ligands, antibodies, or aptamers to selectively capture and detect antigens or nucleic acids [11]. Additionally, their optical, electrical, magnetic, or plasmonic properties allow for signal amplification and real-time monitoring. Nanomaterials with a unique size provide an integrative and portable platform for virus detection, which is suitable for the development of point-of-care (POC) tests. For instance, a nanomaterials-based microbiosensor combined with smart devices has overcome the inconvenience of large equipment and achieved real-time detection. Furthermore, nanoparticles can be used as SERS substrates to develop an analysis model for virus detection, which is a newly developed method with a high diagnosis efficiency.

Therefore, nanomaterials provide innovative solutions for achieving the sensitive and rapid detection of HBV and HCV. Ongoing research in this field aims to improve early diagnostic accuracy and contribute to the better management of these viral infections. Here, we categorized and illustrated the novel approaches for the diagnosis of HBV and HCV, especially those based on nanomaterials (Figure 1).

## 2. Nanomaterials-Based Detection of HBV and HCV

### 2.1. Zero-Dimensional Nanomaterials

#### 2.1.1. Nanoparticles Based Assays

Nanoparticles such as gold, silica, and magnetic nanoparticles can be functionalized with biological components specific to antigens or DNA for detection (Table 1).

Gold nanoparticles (AuNPs): AuNPs are comparatively easy to produce and are suitable for carriers in many biosensors due to their biocompatibility and optical and electronic properties [57]. AuNPs can be conjugated with anti-hepatitis B virus surface antigen (HBsAg) antibodies to create nanogold-HBV-antibody conjugates, which can be applied to an ELISA plate [12]. This method has solved the problem of the low sensitivity of both fluorescence immunoassays and ELISA by human alpha-thrombin (HAT) being labeled on the AuNPs to enhance the fluorescence signals. The limit of detection (LOD) was 10^4^ times lower than that of a conventional fluorescence assay and 10^6^ times lower than that of the conventional ELISA. To detect the possibility of this method for clinical application, the authors found that HBsAg could be detected even after a 1000-fold dilution of patients’ serum samples, which suggested the clinical potential of this highly sensitive AuNP dual labeled probe.

The distinct optical response of AuNPs to electromagnetic radiation leads to an electron oscillation termed surface plasmon resonance (SPR), which is contingent upon the shape and size of a particle and the surroundings. Colorimetric methods based on the SPR of AuNPs can be performed to observe the color change without the need for sophisticated equipment by monitoring the observable color change. Bifunctional polystyrene nanospheres (PSs) modified with goat anti-hepatitis B surface antibodies could be captured and could then catalyze the production of AuNPs [13]. Due to the SPR of AuNPs, the change in the color of the solution can be visibly detected, and the LOD of HBsAg was 0.1 ng/mL when observed with the naked eye, which does not require complicated sample pretreatment or large-scale equipment. AuNPs coated with specific HBV antibodies on a paper strip can be used in the detection of HBV. The results are visible through a color change, which is quick and simple. This method uses very basic equipment for quick diagnostic testing and is suitable in remote areas lacking professional personnel.

Mohammad et al. proposed a cost-effective colorimetric AuNPs-based assay to detect HCV RNA in clinical samples using non-thiolated antisense oligonucleotide [14]. The citrate capped AuNPs could be stabilized and remained red against salt-induced aggregation in HCV-positive samples. Without the requirement of thiol tagging and expensive infrastructure, this method is rapid and cost-effective, which is suitable for application in screening in resource-limited areas. However, the limitation of it is the need for HCV RNA extraction. On the basis of this AuNPs-reliant assay, they also presented a streamlined diagnostic approach for the direct genotyping of globally prevalent HCV genotypes 1 and 3 without PCR by, respectively, designing genotype antisense oligonucleotides [15]. Azzazy et al. used magnetic nanoparticles functionalized with a specific oligonucleotide to extract HCV RNA [16]. The extracted RNA was subjected to react with oligonucleotide sequence targeted HCV RNA in the presence of unmodified cationic AuNPs. In positive samples, the AuNPs were aligned along the phosphate backbone of the RNA, causing aggregation, which resulted in a change in the solution color from red to blue.

Electrochemical biosensors utilizing AuNPs have also been devised to detect HBV. These biosensors rely on a combination of AuNPs coated with oligonucleotides or antibodies and specific DNA fragments or proteins, leading to the generation of amperometric, potentiometric, or impedimetric signals. Wu et al. electrochemically deposited AuNPs on the hemispherical surfaces of electrodes, and a specially designed gene fragment of the target DNA of HBV was immobilized on the nanostructured electrodes as the capture probe [17]. The detection limit was 111 copies/mL. This PCR-free method allowed for the direct detection of HBV DNA from clinical samples without the limitation of conventional assays, such as expensive equipment and a longer detection time. Cui et al. developed AuNPs with both a chemiluminescent (CL) reagent and a catalytic metal complex, and a DNA strand complementary to the target was used as the capture probe for detecting specific DNA sequences from HBV and other pathogens [18]. The bifunctionalized AuNPs enabled the appearance of a well-defined CL signal. Compared to PCR-based methods, this label-free CL method is significantly simpler, faster, and more inexpensive, making it highly suitable for portable POC tests. Double-functionalized AuNPs immobilized a HBsAg-specific aptamer for rapid magnetic separation and amplified the detection signal, therefore allowing for the highly sensitive detection of HBsAg by a specific chemiluminescent aptasensor [19]. The high specificity of the aptamer enabled a high specificity toward the HBV target, even in mixed sera of hepatitis A, B, and C. However, the LOD was only 10 times lower than that of a conventional ELISA assay.

AuNPs amplify the process arising from the high-energy signal of lateral flow assay (LFA) biosensors. The size of AuNPs can be well controlled to have a narrow size distribution, which enhances the colorimetric signals when developing LFA biosensors for HBsAg detection [58]. Recently, the LOD of an AuNPs-based LFA for HBV nucleic acid detection was improved to 10^3^ copies/mL [20]. The enhanced LFA showed great potential for disease diagnosis at the POC. Colloidal gold nanoparticles combined with streptavidin fusion proteins can bind specifically to biotin moieties immobilized on rapid test strips in a dose-responsive manner, which has been utilized to develop a lateral-flow-based rapid test for detecting anti-HCV antibodies in blood samples [21]. Test kits (colloidal gold) for detecting HBsAg and HCV antibodies in plasma have been developed.

Zhang et al. developed an AuNPs probe-based assay to detect HCV core antigen (HCVcAg) by labeling anti-HCVcAg polyclonal antibodies and barcode signal DNA on AuNPs [22]. The LOD of 1 fg/mL was orders of magnitude greater than that of ELISA (2 ng/mL).

Recently, researchers have focused their attention on gold nanorods (AuNRs), which are elongated rod-like gold nanoparticules. AuNRs with exceptional absorption and stronger light scattering properties produce two absorption peaks in the visible and near-infrared regions. Modifying the surface of AuNRs with monoclonal HBsAb in a biosensor enabled the detection of HBsAg based on the local SPR behavior of the AuNRs after antibody modification [23]. AuNRs wrapped with a thin layer of cetyltrimethylammonium bromide and fluorescein were designed in a fluorescence resonance energy transfer system to detect HBV DNA, with a LOD of 15 pmol/L [24]. However, the application of this method required HBV DNA to be extracted from the samples and follow a PCR process.

Metal oxide nanoparticles: these types of metal oxide nanoparticles are more abundant than metal nanoparticles. Metal oxide nanoparticles with various structures such as block, rod, plate, and hollow have brought about many unique properties used for rapid and sensitive virus detection. Zhu et al. developed Cu_2_O hollow microspheres, which can be used to construct a DNA sensor to detect HBV [25]. The excellent selectivity and sensitive response of the device can be attributed to the high specific surface area of the Cu_2_O nanoparticles and their high affinity for the ssDNA probe. Marche functionalized magnetic nanoparticles (MNPs) with anti-HBsAg antibody for the detection of HBsAg by a magnetically localized and wash-free fluorescence immunoassay, which was able to detect HBsAg as low as 5 IU/mL [26]. This was the first successful detection of antigens in plasma without washing steps. The wash-free immunoassay was user-friendly and its diagnostic sensitivity and specificity are suitable to be applied in clinical settings. The simplified process and reduced hands-on time show potential for POC tests. Recently, a variety of metal oxide particles have been used and combined with a variety of other materials to better exert their electrical, optical, and magnetic properties.

Duta et al. reported the fluorescence-based simultaneous dual oligo sensing of genotypes 1 and 3 by employing probes complementary to the target sequences of the HCV genome [27]. Aldehyde-derivatized MNPs enabled the immobilization of the probes and their magnetic removal from MNPs after hybridization with their target, leading to a quantitative analysis of the target. This method allowed for the multiplex oligo sensing of several pathogens simultaneously, saving time and cost.

Silica nanoparticles: silica nanoparticles (NPs) can be combined with a detection probe for HBV DNA detection in a microcantilever platform [28]. Silica NPs are always used to load polymers such as PAA to amplify the signals for the building of ultra-sensitive assays monitoring the change in the absorption intensity of AuNPs at 530nm with a LOD of 3 fM [29]. This study solved the difficulties of controlling the molecular weight of PAA and lowered the polydispersity index (PDI), leading to a bioassay with a better reproducibility.

#### 2.1.2. Quantum-Dot-Based Assays

Quantum dots (QDs) are semiconductor nanocrystals that emit fluorescent light when excited by a light source. In the field of biosensors, QDs have several unique properties, such as a wide excitation spectrum, long fluorescence lifetime, large molar extinction coefficients, exceptional photostability, and tunable emission wavelength. QDs can be functionalized with antibodies specific to HBV antigens, which can bind to the virus and enable its detection. They have been used as labels in HBV detection assays due to their bright and stable fluorescence. As such, QDs have been used widely to detect HBsAg (Figure 2) [30,31], HBV DNA [33], and HBV mutants [32]. This study proposes that QDs can serve as a proficient fluorescent probe for detecting variations in HBV DNA. Moreover, QDs offer a straightforward readout system without requiring complex instruments. The multiplex detection of DNA using multicolor QD nanoprobes is a promising field for further research.

A graphene quantum dot (GQD)-modified electrode with specific-sequence DNA molecules as probes was used by Ye et al. to detect HBV DNA [34]. The probe DNA can bind with HBV DNA instead of GQDs and the peak currents will increase depending on the concentrations of the HBV DNA. Without the need for fluorophore labelling or enzyme amplification, this method is highly convenient and affordable, which makes it easier to fabricate.

### 2.2. One-Dimensional Nanomaterials

#### 2.2.1. Silicon Nanowire-Based Sensors

Nanowires possess a diameter of 10–200 nm and exhibit a length-to-width ratio of at least 1000, so they have been utilized in the detection of HBV. The basic principle is functionalizing the surface of the nanowire with antibodies specifically binding to HBV, and detecting changes in the conductivity of the nanowire. SiNW one-dimensional (1-D) nanomaterial has a large surface-area-to-volume ratio that boosts the analytical sensitivity of bio-molecule detection. Yang fabricated polycrystalline silicon nanowire (SiNW) field-effect transistors and then chemically functionalized them to detect HBsAg at the femto-molar level with an LOD of 100 fg/mL [35]. Priolo demonstrated a SiNW optical biosensor with an LOD of 20 copies/reaction for the genome extracted from blood without PCR. The quenching of quantum confined carriers in SiNWs is used as the detection mechanism [36]. The results achieved in the genome analysis by this method are superior to those obtained with the gold standard real-time PCR technique. The SiNW sensor exhibited an exceptional sensitivity and specificity, offering a convenient detection approach and cost-effective manufacturing process, which are crucial for the advancement of a novel category of genetic point-of-care devices.

#### 2.2.2. DNA Nanowires-Based Sensors

One-dimensional DNA nanotubes or nanowires exhibit distinct properties, including adjustable geometry [31], a high aspect ratio [32], and the efficient and precise anchorage of cargo [34]. Therefore, DNA nanowires (DNW) are attractive tools for constructing biosensors.

Zhang et al. developed a novel detection method for HCVcAg based on DNW and terminal deoxynucleo tidyl transferase (TdT) amplification (Figure 3) [37]. After antibody-conjugated DNA recognized the antigen, the DNA sequence was extended by a robust TdT reaction, and methylene blue-loaded DNWs (MB@DNW) were attached to the sequence as signal labels, which led to an amplified electrochemical signal for ultrasensitive HCVcAg detection. Although this strategy showed usefulness for practical application, the complex preparation of electrodes and disposable modified electrodes might hinder its broad clinical application.

#### 2.2.3. Carbon-Nanotube-Based Sensors

Carbon nanotubes (CNTs) are cylindrical structures made of carbon atoms and have been used as sensors for HBV detection owing to their outstanding structural, electrical, and mechanical properties. CNTs can be functionalized with antibodies specific to HBV antigens, which can bind to the virus and enable its detection. A glassy carbon electrode consisting of amino carbon nanotubes was utilized as an electrochemical immunosensor to detect anti-HBc, with a detection limit of 0.03 ng/mL [38]. Dutra et al. developed another electrochemical immunosensor with a polytyramine (PTy) and CNTs composite, which demonstrated linear responses from 1.0 to 5.0 ng/mL and an LOD of 0.89 ng/mL anti-HBc [39]. The PTy-CNT with high catalytic activity on the electrode surface enabled the detection of the antibody interacting with the immobilized antigens, which was achieved by the electron transfer of the reactive amine groups of PTy. This easily performed and reagentless platform is more suitable for POC tests than electrochemical sensors.

### 2.3. Two-Dimensional Nanomaterials

Two-dimensional (2D) nanomaterials include graphene, transition metal dichalcogenides, black phosphorus, and other emerging materials [59]. Owing to their atomic thinness, 2D materials have unique mechanical, electrical, and optical properties that differ from bulk materials. The interaction between 2D nanomaterials and biological systems is very meaningful, owing to their high surface area and distinct electronic and photothermal characteristics, which have enabled biosensors based on them to detect HBV with a high sensitivity and selectivity. Erdem designed cobalt phthalocyanine-modified pencil graphite electrodes (PGEs) to detect HBV DNA [40]. The detection limit was 2.48 µg/mL in fetal bovine serum medium. PGEs are robust, easy to produce, and can be used as more sensitive protocols compared to glassy carbon electrodes.

Graphene derivatives (such as graphite oxide) have adjustable spectral characteristics and are the preferred materials in the field of optical sensing. Surface chemical modification enables graphene derivatives with different surface functional groups, which can improve their physical and spectral properties to meet specific virus detection needs [60]. Cao et al. synthesized nitrogen- and sulfur-codoped reduced graphene oxide (GO) to detect HBV DNA via fluorescence quenching (Figure 4) [41]. Ma et al. developed a G-quadruplex (GQ)-GO system to accurately quantify the HBV gene with the LOD of the analyte DNA at 0.1 μM, which was the first optimized split GQ-based phosphorescence assay for the HBV gene [42]. GO could significantly reduce the initial luminescence background, which enables a greater increase in the luminescence caused by the analyte.

To achieve the highly sensitive detection of HCV RNA, Fan et al. utilized reduced graphene oxide nanosheets (rGONs) with the hybridization chain reaction (HCR) amplification technique [43]. The weak adsorption to rGONs made the long nanowires produced by target RNA triggering a HCR emit a strong fluorescence. The LOD was 10 fM, which is significantly lower than the typically used GO-based fluorescence approach. This method, with a high selectivity, can discriminate between complementary and mismatched sequences.

Chailapakul et al. reported that single-walled CNTs with Pt nanoparticles deposited on them could be used as substrates for antibodies’ immobilization on an electrode surface, thus boosting the electrochemical sensitivity. It was used to modify paper-based, screen-printed graphene electrodes (SPGE) for the detection of HCVcAg through sensitive, disposable, and inexpensive means, which showed satisfactory results when applied in clinical samples [44]. However, the modified electrode cannot be reused and multiple steps immobilizing the antibody are necessary to achieve high-specificity detection.

### 2.4. Combination of Different Nanomaterials

The effectiveness and applicability of detection methods can be enhanced by utilizing diverse types of nanomaterials, thereby combining their unique properties. In recent years, more and more biosensors have begun to use nanocomposites.

MNPs can be combined with many nanomaterials and these materials therefore play synergistic roles with each other. Fe_3_O_4_ MNPs, efficient biological carriers, can be used as probe collectors, and Rhodamine B loaded-mesoporous SiNPs act as fluorescent capsules and amplifiers, for which the mechanism is capturing and releasing the entrapped dye in the capsules. The fluorescence signal showed a linear relationship with the HBsAg concentration from 6.1 ag/mL to 0.012 ng/mL with an LOD of 5.7 ag/mL [45]. However, the preparation of this probe is time consuming. This limitation might be solved by using already prepared substrates. To detect HBsAg, Samili prepared an electrochemical immunosensor employing Fe_3_O_4_ MNPs with HBs (Ab1) immobilized on it and assembled alkylthiol/G-quadruplex DNA/hemin and antibody on AuNPs as bio-bar-coded nanoparticles [46] (Figure 5a).

GO can also be combined with many nanomaterials. Tajudin et al. immobilized HBcAg onto an AuNPs-decorated rGO nanocomposite as an antigen-functionalized surface to detect anti-HBcAg with an LOD of 3.80 ng/mL [47]. A GO-AuNRs composite was developed for the trace determination of HBsAg in an immunoassay using surface-enhanced Raman spectroscopy (SERS) [48]. AuNRs carrying the SERS probe 2-mercaptopyridine with a high SERS activity enhanced the sensitivity of the biosensor. The antibody on the GO-AuNRs displayed an exceptional specificity towards HBsAg, leading to an excellent selectivity. In another study, GO/Fe_3_O_4_/Prussian blue (PB) nanocomposites and AuNPs were coated on screen-printed electrodes to immobilize the HBsAb and enhance the detection sensitivity of HBsAg [49] (Figure 5b). The decrease in the peak currents of PB was proportional to the concentration of HBsAg captured on the immunosensor. The GO/Fe_3_O_4_/PB nanocomposites functioned as substrates, enhancing the electron transfer and electrochemical redox mediator. GO generated a synergistic effect with the MNPs and AuNPs, which further amplified the signal.

A colorimetric assay based on GQD-silver nanocomposites (GQD/Ag NCs) has been reported to detect HCV RNA [50]. During the catalysis reaction caused by the target, hydrogen peroxide is produced, leading to a color change in the GQD/Ag NCs solution, from light yellow into a colorless silver-ions solution. Simple and sensitive, this retro-transcription-free approach is suitable for HCV screening or expansion to the detection of other RNA viruses. However, the need for sample pretreatment remains a major obstacle to the clinical implementation of this method. In an electrochemical immunosensor, silver nanoparticles (AgNPs) were immobilized onto the SH groups of GQDs through the formation of Ag-S, and anti-HCV was loaded onto the electrode surface via interaction with AgNPs [51]. The working electrode modified by the Ag/GQD-SH nanocomposite exhibited excellent bioactivity and could be utilized for the detection of HCVcAg (Figure 5c). In another electrochemical immunosensor for detecting HCVcAg [52], the electrode was modified using a nanocomposite of graphitized mesoporous carbon-methylene blue (GMCs-MB), while a secondary antibody layer composed of horseradish peroxidase-DNA-coated carboxyl multi-wall carbon nanotubes was used to amplify the signal.

Metal–organic frameworks (MOFs) have emerged as a promising material for developing nucleic-acid-based biosensors due to their distinctive characteristics, such as a low manufacturing cost, large surface area, and controllable porosity [61]. Nevertheless, their limited electrical conductivity might pose a challenge for certain MOFs when designing electrochemical-based biosensors. Sheta et al. constructed a polyaniline@nickel MOF-nanocomposite-based electrochemical biosensor to detect unamplified HCV nucleic acid [53] (Figure 5d). This composite, with a high porosity and excellent electrical conductivity, overcame the low electrical conductivity of traditional MOFs.

Nanomaterials with different morphologies and structures have a large number of active sites and more specific area for catalytic reactions, which have been combined with different nanomaterials for synergistic effects. Sea-urchin-like bimetallic nanoparticles (BMNPs), gold hybrid platinum nanoparticles, and L-cysteine-connected gold-silver nanoparticles have loaded a large number of signal-source molecules. These nanoparticles can be combined with three-dimensional SnO_2_-loaded graphene sheets (GS-SnO_2_-BMNPs) to fabricate sandwich-type immunosensors for detecting HBsAg and HBeAg simultaneously [54]. The square wave voltammetric signal can be enhanced by GS-SnO_2_-BMNPs with an exceptional electrical conductivity and bimetallic synergy. Dong et al. designed a sandwich-type electrochemical immunosensor using Au core and Pd shell nanodendrites (Au@Pd NDs) loaded on amino functionalized molybdenum dioxide nanosheets (MoO_2_ NSs) to detect HBsAg, with an LOD of 3.3 fg/mL [55]. Because of the plentiful catalytic activity sites offered by the surface dendrite structure, the catalytic reduction ability of the Au@Pd NDs surpassed that of single gold and palladium nanoparticles. MoO_2_ NS, with a two-dimensional structure, was used to prevent the self-aggregation of Au@Pd nanodendrite materials. This ultrasensitive method showed an excellent accuracy and stability in the detection of serum samples.

Flower-like gold nanoparticles (AuNFs) also possess an expanded specific surface area with abundant active sites and a stronger local electromagnetic field enhancement effect, facilitating enhanced catalytic activity [62,63]. Chitosan, luminol, and Co^2+^-functionalized AuNFs with strong CL properties have been reported for the label-free sensing of the HCVcAg [56]. The approach holds universal potential for advancing the ultrasensitive detection of various proteins in early disease diagnosis.

Combinations of nanomaterials have allowed for the collaboration of different elements and the usage of their special characteristics for an innovative platform.

### 2.5. Application of Smart Phones and Machine Learning in HBV and HCV Detection

Bulky, expensive, laboratory-based instruments are not suitable for POC applications. With the rise of nanomaterials in biosensors, an electrochemical biosensor based on smart phones has been developed, which can realize electrochemical detection, data analysis, and transmission through mobile phone applications. It possesses the benefits of portability, speed, and an economical and easy operation. Chailapakul et al. created a smartphone-controlled electrochemical sensor based on Near Field Communication to quantify HBsAg, with AuNPs-modified SPGE [64]. The LOD for HBsAg was 0.17 μg/mL. The results were confirmed by traditional immunoassays. This electrochemical immunosensor showed promise as an alternative means for creating portable, user-friendly, highly accurate, and specific tools.

The great improvements in detection level and efficiency put forward higher requirements for the rapid analysis of detection data, and virus detection methods based on machine learning have gradually attracted more attention. It is imperative to develop models utilizing advanced approaches such as deep neural networks in order to accurately discern the subtle variations between healthy and infected samples.

For example, Raman spectroscopy is capable of identifying molecules through the emission of inelastically scattered light that arises from the intra-bond vibrational mode. Analyzing Raman spectroscopic images to accurately identify the HBV disease presents a challenge for pathologists. Tahir et al. suggested an innovative approach through the application of deep neural networks, which was based on using pre-trained convolutional neural networks ResNet101 on a genuine dataset of HBV-infected blood samples [65]. Furthermore, this automated approach with a high performance can be developed for other diseases.

Compared to Raman Spectroscopy, SERS offers advantages, particularly in detecting trace concentrations of proteins [66] and RNAs [67]. SERS has been effectively employed for the diagnosis of infectious diseases with an exceptional sensitivity and specificity [68,69]. Several studies have performed AgNPs-based SERS analyses of HCV-positive samples and the viral RNA extracted from them, comparing these with samples from healthy donors. A Principal Component Analysis was performed and a Partial Least Square Regression Analysis model was developed for predicting HCV viral load with a 99% accuracy [70,71,72]. The LOD for HCV RNA samples via SERS is 2.55 log IU/mL. This method might be an alternative technique for the qualitative and quantitative detection of HCV.

## 3. Other New Detection Methods

Technologies based on the CRISPR-associated protein (Cas) system are expected as a new generation of nucleic acid detection methods for achieving rapid, accurate, and portable nucleic acid diagnosis [73]. Ren et al. established a new CRISPR-Cas13a-based assay based on rolling circle amplification and PCR to detect covalently closed circular DNA (cccDNA) in liver tissue, which can detect 1 copy/μL HBV cccDNA after amplification [74]. Ren et al. also established a colloidal gold test strip for detecting HBV DNA in low-level viremia patients, which utilized CRISPR/Cas13a combined with recombinase-aided amplification technology [75]. With the sensitivity of 10^1^ copies/μL and a specificity of 100%, the strip detection identified clinical samples effectively.

## 4. Conclusions

The review summarized the modern approaches based on the various nanomaterials that have been used for the diagnosis of HBV and HCV. In the future, more highly sensitive novel approaches are expected. On the other hand, low- or middle-income countries require screening techniques that are simpler, easy to access, and affordable. Overall, the use of nanomaterials, especially in POC testing, enables the improvement of sensitivity, specificity, and speed for HBV and HCV diagnosis, which can lead to earlier interventions and better outcomes. Nanomaterials offer a hopeful approach to improving the diagnosis and treatment of HBV and HCV, which remains a major global health challenge.

## Figures and Tables

**Figure 1 molecules-28-07201-f001:**
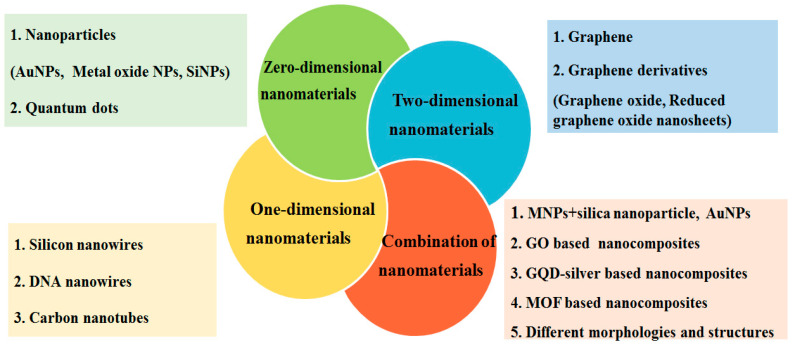
Nanomaterial-based approaches for detection of HBV and HCV. AuNPs: gold nanoparticles; MNPs: magnetic nanoparticles; SiNPs: silica nanoparticle; QDs: quantum dots; GO: graphene oxide; and MOFs: metal–organic frameworks.

**Figure 2 molecules-28-07201-f002:**
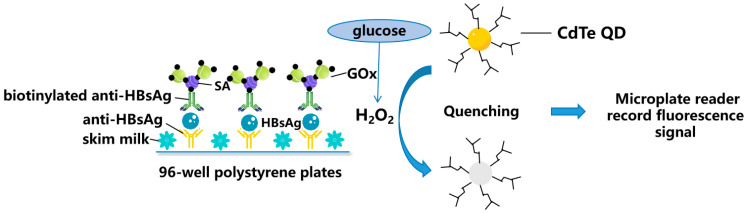
Schematic representation for zero-dimensional nanomaterials detecting HBV simplified from [31]: Fluorescence enzyme-linked immunosorbent assay based on glucose oxidase (GOx)-mediated fluorescence quenching of quantum dots (QDs) for highly sensitive detection of Hepatitis B. SA: Streptavidin; and CdTe: Cadmium Telluride.

**Figure 3 molecules-28-07201-f003:**
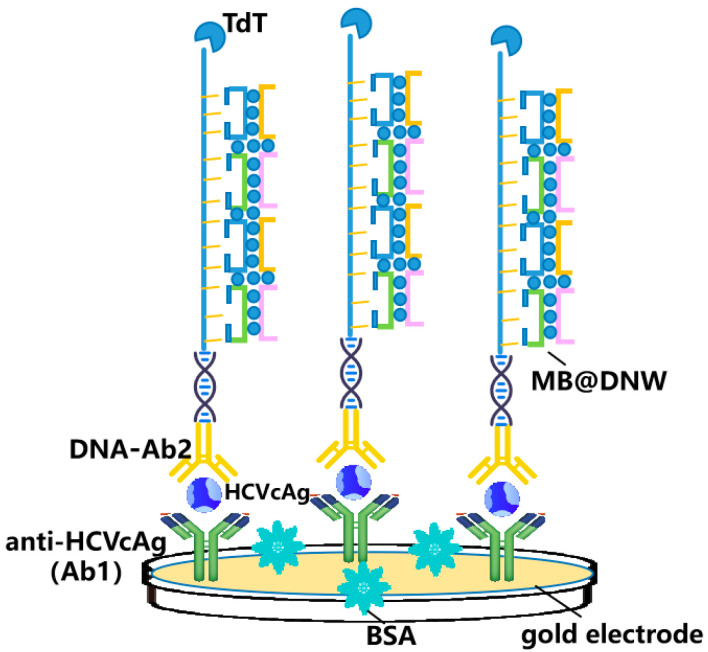
Schematic representation for one-dimensional nanomaterials detecting HCV simplified from [37]: DNA nanowires (DNW)-based assay for detection of HCVcAg.

**Figure 4 molecules-28-07201-f004:**
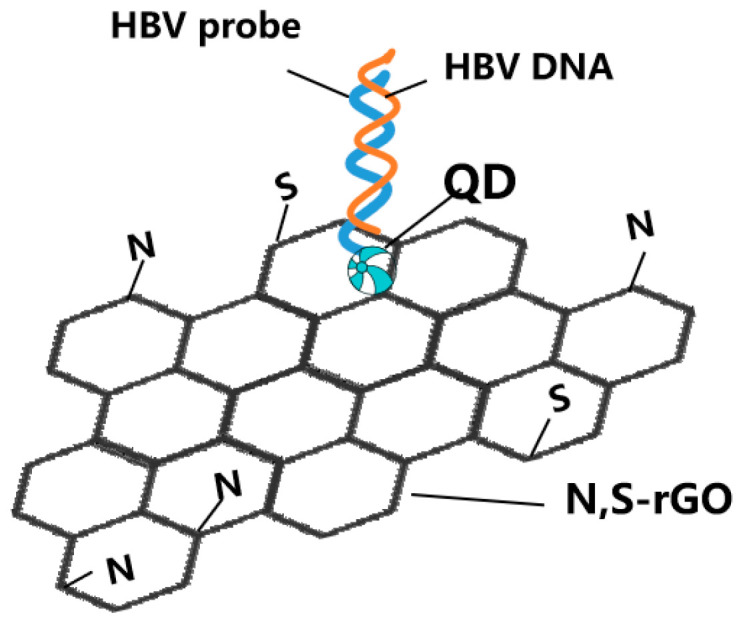
Schematic representation for two-dimensional nanomaterials detecting HBV simplified from [41]: nitrogen- and sulfur-codoped reduced graphene oxide (N,S-rGO)-based assay for detection of HBV DNA.

**Figure 5 molecules-28-07201-f005:**
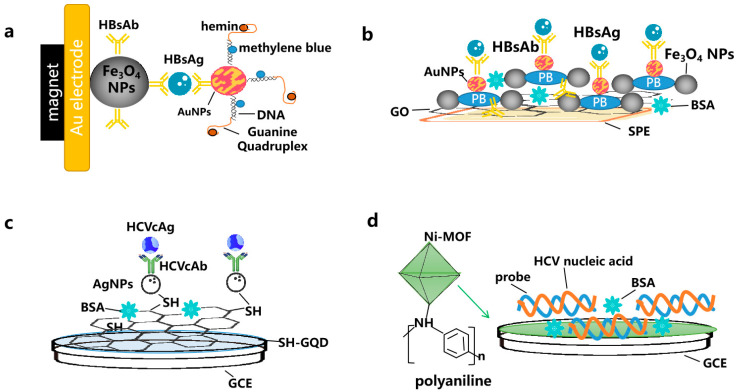
Schematic representation for combination of different nanomaterials detecting HBV and HCV simplified from [46,49,51,53]: (**a**) Electrochemical immunosensor employing Fe_3_O_4_ magnetic nanoparticles (MNPs) and gold nanoparticles (AuNPs) to detect hepatitis B virus surface antigen (HBsAg); (**b**) screen-printed electrodes (SPE) with GO/Fe_3_O_4_/Prussian blue (PB) nanocomposites and AuNPs coated on them to detect the HBsAg; (**c**) glassy carbon electrode (GCE) modified by Ag/graphene quantum dot (GQD)-SH nanocomposite for the detection of hepatitis C virus central antigen (HCVcAg); and (**d**) A polyaniline@nickel metal–organic frameworks (MOFs) nanocomposite-based electrochemical biosensor to detect unamplified HCV nucleic acid.

**Table 1 molecules-28-07201-t001:** Recent advances in nanomaterial-based applications for detection of HBV and HCV.

Structure	Nanomaterial Use	Target	Detection Method	Lower Limit of Detection	Linear Range	Ref.
	Anti-HBs and HAT-coated AuNPs	HBsAg	Fluorescence ELISA	5 × 10^−4^ IU/mL	N.A.	[12]
	Bifunctional PSs and AuNPs	HBsAg	Color change of solution due to SPR	0.1 ng/mL (naked eye) 0.01 ng/mL (instrumentation)	0.01–10 ng/mL	[13]
	AuNPs	HCV RNA	Colorimetric, UV-visible spectrophotometer	100 IU/mL	N.A.	[14]
		HCV genotyping		100 IU mL	N.A.	[15]
	AuNPs and MNPs	HCV RNA	Colorimetric	15 IU/ml	N.A.	[16]
	AuNPs	HBV DNA	EIS	111 copies/mL	10^2^–10^5.1^ copies/mL	[17]
	Bifunctionalized AuNPs	HBV DNA	Chemiluminescence detection	5.9 × 10^−12^ M	N.A.	[18]
Nanoparticle	Au@Fe_3_O_4_@SiO_2_NPs	HBsAg	Chemiluminescent aptasensor	0.05 ng/mL	1–225 ng/mL	[19]
	AuNPs	HBV DNA	Lateral flow assay	10^3^ copies/mL	N.A.	[20]
	AuNPs with streptavidin fusion proteins	HCV antibodies	Lateral flow assay	0.048 ng/mL	N.A.	[21]
	AuNPs	HCVcAg	Fluorescence PCR	1 fg/ml	N.A.	[22]
	AuNRs	HBsAg	Direct detection of SPR peak	0.1 IU/mL	0.01–1 IU/mL	[23]
	AuNRs	HBV DNA	FRET	15 pmol/L	0.045–6.0 nmol/L	[24]
	Cu_2_O nanoparticles	HBV DNA	Electrochemical	1.0 × 10^−10^ mol/L	1 × 10^−10^–1 × 10^−6^ mol/L	[25]
	MNPs	HBsAg	Magnetically localized fluorescence immunoassay	5 IU/mL	N.A.	[26]
	Aldehyde-derivatized MNPs	HCV genotypes 1 and 3	Fluorescence spectroscopic measurements	10–15 nM	1–100 nM	[27]
	Silica NPs	HBV DNA	Resonant frequency shifts	2.3 × 10^−15^ M	23.1 fM–2.3 nM	[28]
	Silica NPs	HBV DNA	Electrochemical	3 fM	10–100 fM	[29]
	QDs nanobeads	HBsAg	Fluorescent immunoassay	0.078 ng	N.A.	[30]
	Mercaptopropionic acid-modified cadmium telluride QDs	HBsAg	Fluorescence quenching	1.16 pg/mL	47–380 pg/mL and 0.75–12.12 ng/mL	[31]
QDs	QDs	HBV mutants	Fluorescence microscopy	10^3^ IU/mL	N.A.	[32]
	QDs	HBV DNA	FRET	1.5 nmol/L	2.5–30 nmol/L	[33]
	Graphene QD	HBV DNA	DPV	1 nM	10–500 nM	[34]
One-dimensional materials	SiNW	HBsAg and HBx	Electrochemical	100 fg/mL	N.A.	[35]
	SiNW	HBV DNA	Quenching	20 copies/reaction	N.A.	[36]
	Methylene blue-loaded DNA nanowires	HCVcAg	DPV and EIS	32 fg/mL	0.1–312.5 pg/mL	[37]
	Amino CNTs	HBcAb	SWV	0.03 ng/mL	0.03–6 ng/mL	[38]
	Polytyramine and CNTs	HBcAb	SWV	0.89 ng/mL	1.0–5.0 ng/mL	[39]
Two-dimensional materials	Pencil graphite electrodes	HBV DNA	EIS and CV	2.48 µg/mL	5–30 μg/mL	[40]
	N,S-rGO	HBV DNA	Fluorescence quenching	2.4 nmol/L	5–100 nmol/L	[41]
	G-quadruplex-GO system	HBV DNA	Split phosphorescence amplification assay	0.1 μM	2–300 nM	[42]
	rGO nanosheets	HCV RNA	Fluorescence quenching	10 fM	10 fM–100 pM	[43]
	Pt single-walled CNT-modified graphene electrode	HCVcAg	DPV	0.015 pg/mL	0.05–1000 pg/mL	[44]
	Fe_3_O_4_ MNP and Rhodamine B-mesoporous silica nanoparticle	HBsAg	Fluorescence of solutions	5.7 ag/mL	6.1 ag/mL–0.012 ng/mL	[45]
	Fe_3_O_4_ MNP and AuNPs	HBsAg	SWV	0.19 ng/μL	0.3–1000 ng/μL	[46]
	rGO-en-AuNPs	HBcAg	EIS	0.19 pg/mL	3.91–125.00 ng/mL	[47]
Nanocomposite	GO-GNRs	HBsAg	Surface-enhanced Raman spectroscopy	0.05 pg/mL	1–1000 pg/mL	[48]
	GO/Fe_3_O_4_/ Prussian blue	HBsAg	Electrochemical (CV, DPV, and EIS)	0.166 pg/mL	0.5–200 ng/mL	[49]
	GQD-silver nanocomposites	HCV RNA	Colorimetric	24.84 pM	25–500 pM	[50]
	AgNPs/Thio-GQDs	HCVcAg	DPV	3 fg/mL	0.05 pg/mL–60 ng/mL	[51]
	Graphitized mesoporous carbon-methylene blue carboxyl multi-wall carbon nanotubes	HCVcAg	SWV	0.01 pg/mL	0.25–300 pg/mL	[52]
	polyaniline@nickel metal–organic framework	HCV nucleic acid	Electrochemical (CV and EIS)	0.75 fM	1 fM–100 nM	[53]
	Graphene sheets -SnO_2_- bimetallic MNPs	HBsAg	SWV	4.67 pg/mL	0.01–100 ng/mL	[54]
		HBeAg	SWV	4.68 pg/mL	0.01–100 ng/mL	
	Au@Pd nanodendrites /NH_2_-MoO_2_ nanosheets	HBsAg	Electrochemical (CV and EIS)	3.3 fg/mL	10–100 ng/mL	[55]
	Co^2+^/Chitosan/Luminol/AuNFs	HCVcAg	Electrochemical	0.16 ng/mL	0.50 ng/mL–1.00 μg/mL	[56]

ELISA: enzyme-linked immunosorbent assay; NPs: nanoparticle; AuNPs: gold nanoparticles; HAT: human alpha-thrombin; HBsAg: hepatitis B virus surface antigen; PSs: polystyrene nanospheres; SPR: surface plasmon resonance; HCVcAg: HCV core antigen; PCR: polymerase chain reaction; EIS: electrochemical impedance spectroscopy; AuNRs: gold nanorods; FRET: fluorescence resonance energy transfer; MNPs: magnetic nanoparticles; QDs: quantum dots; DPV: differential pulse voltammetry; SiNW: silicon nanowire; CNTs: carbon nanotubes; HBcAb: hepatitis B core antibody; SWV: square wave voltammetry; CV: cyclic voltammetry; GO: graphene oxide; AgNPs: silver nanoparticles; HBeAg: hepatitis B virus e antigen; and N.A.: non applicable. AuNFs: flower-like gold nanoparticles.

## Data Availability

Not applicable.

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
