# Peer review of "The Advancement of Nanomaterials for the Detection of Hepatitis B Virus and Hepatitis C Virus"

_molecules, 2023, doi:10.3390/molecules28207201_

Round 1
Reviewer 1 Report
This article reviews nanomaterial-based approaches used for diagnosis of HBV and HCV. The literature cited are latest, but this article should be reconsidered after some issues being addressed.
1. This article reviews many nanomaterial-based methods and states their performance simply, but I think it’s too simple. The authors should give the advantages and disadvantages, as well as suitable analysis for each method, which are important for supporting the further research.
2. What real applications have been achieved by these methods? What occasions these approaches are suitable for? I think the related information should be indicated.
3. What does the cccDNA mean in the main text. It should be noted.
Author Response
Dear professor:
Thank you for your precious time! Please see the attachment.
Best wishes!

Reviewer 2 Report
This review reports on the advancement of nanomaterials for the detection of HBV and HCV. Overall, the article structure is reasonable. The types of nanomaterials are rounded. I can recommend it for publication in Molecules after addressing the following questions:
1. The innovation, necessity and feasibility of nanomaterials for HBV and HCV detection should be mentioned in the Introduction.
2. For some examples, relevant curves and figures should be provided, such as the examples in Sections 2.1 and 2.3.
3. The relationship between Section 2.5 and the others should be demonstrated in the Introduction.
4. Have nanomaterials been used practically in medical detection?
5. The limitations and possible solutions of nanomaterials for HBV and HCV detection need to be discussed at the end of Sections 2.1–2.4.
Author Response

(The authors gave the same response as above.)

Reviewer 3 Report
This manuscript provides a helpful review of how varied types of nanomaterials have been employed to offer new detection capabilities for the B and C virus forms of hepatitis. After a short Introduction section, the authors present the centerpiece of the review in the form of a table (Table 1). The table contains entries representing nanomaterials-based detection work from 45 references. Sections of the table are divided by materials dimensionality (0, 1, 2) and there is also a category for composite nanomaterials. The sections of the paper are organized along these same divisions and each has text consisting of a running list of very concise descriptions of the referenced work.
In my opinion, the review is generally acceptable since the compilation of referenced work in Table 1 offers sensor researchers a useful resource on published work, particularly as the field will continue to move forward. There are some points on which this review could be made easier to read and more substantial. First, there are a number of grammatical errors throughout the text, and I offer a significant list of suggested and needed modifications below (by line number). I will note too that as a review, it includes only two figures - one a simple introductory schematic for the main materials categories of Table 1, and the other offering some adapted graphics to illustrate several nanocomposite detection examples. In my opinion a more complete review would endeavor to include a somewhat longer Introduction, some overarching conceptual aspects (supported by additional instructive figures) which are critical to the role of nanomaterials in hepatitis sensing, as well as more figures to enrich the 0, 1 and 2 dimensional work included in the review (in the style of the included Figure 2 for nanocomposites).
I consider a substantial majority of my listed wording changes as necessary for making the manuscript more readable. My other suggestions would expand the manuscript in important ways to make it a more impressive “Review”.
Suggested Useful Wording Changes/Improvements (by line number: Lxyz)
L9 liver damage
L13 Here we categorize
L18 people living with it. It causes
L19 1.34 million deaths per year (on average?) …. mostly from heapatitus B
L20 relevant diseases
L27 HBV and HCV infections
L36 Here we categorize
L37 illustrate novel approaches for diagnosis of HBV and HCV, especially those based on nanomaterials.
L38 delete this line
L40 Nanomaterials-based approaches
Reference 23 in Table 1: Cu2O {subscript needed here, and throughout the paper}
L52-53 {possible confusion with SiNPs meaning silica nanoparticles, and SiNW meaning silicon nanowire}
L61 color indicates
L65 termed surface plasmon resonance
L75 are visible through
L88 aggregation which resulted
L89 AuNPs have also been devised
L93 on the hemispherical surfaceproteins, and imaging
L103 AuNPs amplify process arising from high-energy
L104 well controlled to have a narrow size distribution which enhances colorimetric
L107 for disease diagnosis
L108 the point-of-care
L109 on rapid test strips
L114 LOD of 1 fg/ml was orders of magnitude greater
L116 (AuNRs), which are
L119 HBsAb in a biosensor
L123 DNA, with a LOD of 15 pmol/L
L125 {I do not know what the authors are trying to convey here}
L127 Cu2O
L129 Cu2O
L130 functionalized magnetic nanoparticles
L131 by a magnetically localized
L138 with their target, leading
L140 {see L52-53 comment above} combined with a detection
L143 on them were
L144 with a LOD
L148 such as a wide
L149 photostability, and tunable emission
L154 with specific-sequence DNA
L161 utilized to advantage in the biomolecular expression at the single-cell
L162 nanowire with antibodies
L165 fabricated polycrystalline silicon
L169 as the detection
L174 tools for constructing biosensors.
L177 recognized the antigen, the DNA sequence
L182 Carbon nanotubes (CNTs)
L186 utilized as an electrochemical
L196 which has enabled biosensors based on them to detect
L212 triggering HCR to emit
L217 on them could be used
L225nanomaterials, thereby combining …. properties. In recent years
L226 more biosensors have begun to use nanocomposites.
L227 MNPs can be ….. nanomaterials, and these materials therefore
L229 silica nanoparticles act as fluorescent capsules
L230 and amplifiers, for which
L235 AuNPs as a bio-barcoded nanoparticle
Figure 2 {has a red and yellow object for which the “AuNPs” label is difficult to read}
L256 on them to detect HBsAg;
L261 onto an AuNPs-decorated
L262 with a LOD
L275 from light yellow into a colorless silver-ion solution.
L278 modified by the Ag/GQD-SH
L280 In another
L284 amplify the signal.
L286 biosensors due to their distinctive characteristics such as
L288 pose a challenge
L291 This composite
L295-299 {the content here is unclear to me - and should be revised to make the meaning clear}
L302 using an Au core
L305 offered by the surface
L315 biosensors, an electrochemical
L325 data, and virus detection methods
L326 machine learning have gradually
L328 subtle variations
L330 emission of inelastically scattered light that arises
L336 diseases.
L340 AgNPs-based SERS ….. positive samples,
L341 from it, compared
L342 and a Partial Least
L348 CRISPR-associated protein
L358 This review has summarized modern approaches
L359 which have been used for diagnosis
L362 nanomaterials, especially in POC testing, enables
L433 Cu2O
The manuscript would be much easier to read with the corrections and changes I have suggested (see comments above) from Line 9 to Line 433.
Author Response

(The authors gave the same response as above.)

Round 2
Reviewer 1 Report
The authors respond all the questions, and the quality of this article has been improved. I aggree its publication after solving the following minor errors.The “figure” in line 193 and line 241 should be corrected as “Figure”. I suggest the authors check the article thoroughly and correct spelling errors carefully.
Reviewer 2 Report
This review has been improved much after revision. I think it can be accepted at current state.